# Temporal Leptin to Determine Cardiovascular and Metabolic Fate throughout the Life

**DOI:** 10.3390/nu12113256

**Published:** 2020-10-24

**Authors:** Jae Geun Kim, Byung Ju Lee, Jin Kwon Jeong

**Affiliations:** 1Division of Life Sciences, College of Life Sciences and Bioengineering, Incheon National University, Incheon 22012, Korea; jgkim@inu.ac.kr; 2Institute for New Drug Development, Division of Life Sciences, Incheon National University, Incheon 22012, Korea; 3Department of Biological Sciences, College of Natural Sciences, University of Ulsan, Ulsan 44610, Korea; 4Department of Pharmacology and Physiology, School of Medicine & Health Sciences, The George Washington University, Washington, DC 20037, USA

**Keywords:** leptin, central nervous system, cardiometabolic fate, body weight, adiposity

## Abstract

Leptin links peripheral adiposity and the central nervous system (CNS) to regulate cardiometabolic physiology. Within the CNS, leptin receptor-expressing cells are a counterpart to circulating leptin, and leptin receptor-mediated neural networks modulate the output of neuroendocrine and sympathetic nervous activity to balance cardiometabolic homeostasis. Therefore, disrupted CNS leptin signaling is directly implicated in the development of metabolic diseases, such as hypertension, obesity, and type 2 diabetes. Independently, maternal leptin also plays a central role in the development and growth of the infant during gestation. Accumulating evidence points to the dynamic maternal leptin environment as a predictor of cardiometabolic fate in their offspring as it is directly associated with infant metabolic parameters at birth. In postnatal life, the degree of serum leptin is representative of the level of body adiposity/weight, a driving factor for cardiometabolic alterations, and therefore, the levels of blood leptin through the CNS mechanism, in a large part, are a strong determinant for future cardiometabolic fate. The current review focuses on highlighting and discussing recent updates for temporal dissection of leptin-associated programing of future cardiometabolic fate throughout the entire life.

## 1. Introduction

Leptin, one of the most extensively characterized adipokines produced mainly from the white adipose tissue, travels the entire body through the bloodstream and triggers leptin receptor-associated signaling to regulate a broad spectrum of whole body physiological homeostasis, including metabolic and cardiovascular systems. Importantly, circulating leptin can directly access target regions within the central nervous system (CNS) by passing through the blood–brain barrier (BBB) [1]. Then, leptin modulates molecular, cellular, and synaptic networks within the CNS to regulate metabolic parameters, including appetite behavior, energy expenditure, peripheral adipose and glucose metabolism, as well as cardiovascular tone, such as blood pressure and sympathetic nervous activity (SNA) [2,3,4,5,6]. Disruption in central leptin signaling, such as leptin resistance, leads to the development of metabolic diseases, such as obesity, type 2 diabetes, and hypertension, all of which are leading causes of death in the modern world. Therefore, leptin is the key molecule that connects the peripheral and central systems in the regulation of cardiometabolic physiology.

Developmental environment is a considerable factor that influences physiological outcome in later life, and a beneficial role of the physiological intake of breastmilk leptin in metabolic programming has been previously reviewed [7]. On the other hand, alterations in the developmental and/or earlier environment are directly linked to pathophysiological development at a later time [8]. For example, impairment in leptin signaling in early life results in the development of permanent cardiometabolic abnormality [9,10,11,12]. Both animals and humans with genetic deletions or mutations of either leptin or leptin receptor develop insulin resistance and a severe obese phenotype in postnatal life [9,10,11,12,13,14,15,16]. Furthermore, recovery of leptin signaling in the postnatal period of leptin signaling-deficient animals failed to reverse most of the metabolic abnormalities, indicating the existence of developmental time-sensitive and leptin-dependent programming for cardiometabolic functioning in later life [11]. In fact, accumulating evidence points to leptin as a key regulator for fetal development, neonate weight, size, adiposity, and head circumference in humans, all of which are potentially indicative of cardiometabolic state at a later stage [17,18]. In addition, extensive investigation has indicated a critical role for leptin in adulthood as well to set future cardiometabolic fate [19,20,21,22]. Therefore, leptin appears to play a role in temporally regulating cardiovascular and metabolic fate throughout life. In this review, we will provide a more recent update for early (from gestation to childhood) and adulthood leptin as a predictor of future cardiovascular and metabolic fate. 

## 2. Early Leptin for Future Cardiometabolic Output

### 2.1. Maternal and Cord Leptin during Gestation 

During the typical gestational period, maternal serum leptin concentration continually increases following gestational progress until it reaches the peak level in later stages (approximately 37 ± 28 ng/mL; [23]), then dramatically drops to return back to pre-pregnant levels after delivery, as shown in Figure 1 [23,24,25,26]. Leptin induction during gestation is driven by (1) an enhanced leptin production correlated with maternal body weight gain, (2) an extra production of leptin from the placenta, and (3) a development of insulin and leptin resistance due to impaired leptin signaling in the CNS. This gestational leptin is closely linked with pregnant-specific glucose metabolism in women that supports the growth of an infant [26,27,28,29]. Investigation using rodent models further demonstrated that this gestational metabolic adaptation is necessary for the normal development of offspring. Ablation of maternal leptin receptor downstream signaling induced a reduction in brain mass that was paralleled with lowered hypothalamic proopiomelanocortin synapses in their offspring [30]. These investigations indicate an effect of maternal metabolic adaptation during gestation on leptin-dependent brain development and function to preserve metabolic homeostasis in the offspring. Importantly, the temporal leptin rhythm during gestation, especially in the early stages, has been suggested to play an important role in fetal development. Additionally, temporal leptin rhythm influences the physiological outcome of the neonate, such as size, weight, and adiposity, all of which are potential contributing factors for future metabolic and cardiovascular output [18,23,31,32,33]. 

For example, the level of serum leptin is positively correlated with the mother’s body mass index (BMI) during typical pregnancy conditions, and fetuses exposed to higher leptin tend to have higher birth weights [23,33]. Oppositely, reduced maternal leptin levels are significantly correlated with lower birth weight in the neonate [26,33,34]. Therefore, maternal weight gain, leptin, and neonate birth weight are positively intercorrelated during normal pregnancy. However, this positive relationship becomes less defined when pregnant women are obese or diabetic [26,32,33]. Again, circulating leptin level in healthy pregnant women is higher than that in non-pregnant females (leptin concentration in healthy adult women is reported around 21.9 ng/mL; [35]), but maternal leptin levels are even higher in women with gestational diabetes mellitus. This abnormal high level of maternal leptin is also positively correlated with maternal insulin resistance [36], and offspring exposed to maternal obesity and/or gestational diabetes mellitus are either growth restricted, normal, or overweight [26,36]. Maternal nutritional imbalances as a risk factor for offspring obesity have been demonstrated not only in humans, but also in rodents [37,38,39]. In rats, maternal high-fat diet before and during the gestational period increased the risk of metabolic disease development in both sexes of their offspring [37]. The increased risk is mediated through sex-specific genetic alterations in downstream molecules of leptin signaling, which impairs hypothalamic leptin signaling prior to the development of metabolic diseases, and results in the onset of metabolic diseases [37]. Importantly, central leptin resistance and degree of metabolic impairment, including body adiposity, and disruption in glucose metabolism in offspring became stronger when the mother was continuously treated with high-fat during the suckling period [38], with an irreversible disruption of hypothalamic leptin mechanism in the offspring [39]. These investigations indicate the necessity for normal maternal nutritional balance and leptin level for healthy neural development and metabolic homeostasis in their offspring. It is also worth noting that leptin is also present in the amniotic fluid in the late gestation period [40]. The amniotic fluid leptin, probably originating from the placenta, could be absorbed into the fetus by swallowing, and may also have been implicated in the programming of fetus cardiometabolic health, and therefore, detailed investigations are permitted.

The genomic and molecular profiles of cord blood, involving differential DNA methylation activity, hormone levels (e.g., insulin, adiponectin, and leptin), and amino acid levels (e.g., branched chain amino acids, aromatic amino acids, and acylcarnitines) are directly implicated in fetal development and fat mass gain [41,42]. Furthermore, these profiles are suggested to be associated with developmental programing in later life [41,42]. Because the degree of cord leptin is positively correlated with maternal leptin levels [18,23,36,43], and representative of fetal growth and newborn weight, adiposity, and BMI during pregnancy [17,18,23,26,36,43,44,45,46], the levels of cord leptin (approximately 15 ± 16 ng/mL; [23]) are considered to be a key predictor of childhood cardiovascular and metabolic state. Cord leptin levels can be linked with birth weight of neonates from women with gestational obesity and/or diabetes mellitus as well [36,47,48]. Together, the degree of leptin concentration in the maternal serum and cord blood could be an indicator for neonate birth weight and adiposity.

The degree of cord leptin is associated not only with the birth weight, but also with postnatal body weight gain [45,46,49,50,51]. While high cord leptin mediates a gradual gain of infant body weight and adiposity [45,51], low neonatal leptin results in rapid body weight gain during the early postnatal period, making low neonatal leptin a strong risk factor for childhood obesity [50]. This negative correlation between cord leptin and infant body weight gain is sustained in early childhood, up to the age of 4, then switches to a positive relationship in later childhood and adolescence [44,46]. In addition to the cord leptin, birth weight itself is a strong programming factor for an individual’s metabolic condition in later life as well. For example, higher birth weight is strongly associated with the development of an obese phenotype in both childhood and adolescence [52]. On the other hand, both high and low birth weights are positively linked with the development of type 2 diabetes mellitus, producing either a U- or J-shaped correlation [53]. 

In addition to metabolic programming, birth weight is also directly associated with future cardiovascular parameters, although the correlations between birth weight and the risk of cardiovascular disease (CVD) are inconsistent between studies. While some investigations recognized a U-shaped association (increased risk of CVD with both lower and higher birth weight), other studies revealed an inverse-type (high risk of CVD only with lower birth weight), presumably due to variability in parameters for low and/or high birth weight [53,54,55,56]. Nevertheless, it appeared to be consistent that a lower birth weight is significantly associated with a high risk of CVD in later life [53,55,56]. In support of this finding, a recent meta-analysis using worldwide information examining the interplay of birth weight and cardiovascular parameters demonstrates a novel J- or inverse-shaped pattern of association between birth weight and the risk of CVD (in this analysis, CVD includes coronary heart disease, stroke, and myocardial infarction), indicating a clear association of the risk of CVD with a low birth weight in both sexes [53]. Importantly, a high birth weight in females, but not in males, is also positively correlated with the risk of CVD, suggesting an interaction of the sex effect on the birth weight–CVD relationship [53]. This investigation also recognized a clear inverse-shaped relationship between birth weight and blood pressure in both sexes, and a 1 kg decrease in birth weight is correlated with about 23% increased risk of hypertension in later life [53]. However, this meta-analysis did not examine maternal effects during gestation. To compensate for this gap, another group performed a meta-analysis to investigate the association between gestational diabetes mellitus and the risk of CVD in children exposed to gestational diabetes mellitus [57]. They observed that systolic blood pressure, but not diastolic blood pressure, is increased in children exposed to maternal gestational diabetes mellitus compared to those from an otherwise healthy pregnancy [57]. Not only is leptin in maternal serum and cord blood closely associated with the birth weight of neonates, as aforementioned, but it also has a further role in affecting cardiometabolic fate in later life.

### 2.2. Postnatal Leptin

After birth, leptin levels in newborns (approximately 2.8 ng/mL; [58]) are crucial for postnatal developmental progress. In humans, the serum concentration of leptin begins to decrease during the first week of life (as low as 0.2 ng/mL; [59]) in parallel with a reduction in body weight [59,60]. This leptin–body weight reduction is linked with the growth of infants during their first year of life [59]. Overall, serum leptin stays at low levels through childhood, then, surges before the onset of puberty, as shown in Figure 1 [61]. In rodents, however, the surge in leptin occurs in the first two weeks of life to establish CNS synaptic organization, including the hypothalamic neural network responsible for reproduction and metabolic regulation, via its neurotrophic action [59,62,63,64,65,66]. Early leptin intake has been considered beneficial for cardiometabolic outcomes in later life [7]. For example, breastfeeding is a continuous source of external leptin delivery from mothers to their offspring, although it is dependent on the mother’s physiological status during the suckling period, and the breastfed-mediated high serum leptin levels in the offspring have been considered to play a role in the programming of long-lasting cardiometabolic homeostasis [7]. On the other hand, one investigation has recognized that impairment in insulin sensitivity and glucose metabolism occurs when external leptin is administered in rodents during the neonatal period [67]. Therefore, the effects of leptin on future cardiometabolic programming could be dependent on multiple factors, including the dose and way of administration. As aforementioned, white adipose tissue is the main source of leptin production, and serum leptin concentration follows body adiposity during infancy and childhood [68,69]. Therefore, these investigations indicate a role for leptin in bridging peripheral adiposity with the development and function of the neural circuitry for metabolic regulation during early life. 

Alteration in serum leptin concentration (e.g., leptin deficiency or elevated leptin level) is one of the strongest single contributors causing childhood early onset obesity [70]. Children with mutations in genes encoding leptin or leptin receptors develop childhood monogenic obesity with higher BMI than age-matched, obese children without leptin deficiency [71]. In parallel, a longitudinal follow-up study with serum leptin measurements through prepubertal childhood (between age 0 and 9 years) recognized a positive relation between serum leptin level and BMI [72]. Interestingly, this study also demonstrated that leptin levels at 2 years of age can predict future leptin levels at the age of 5 and 9 years. As a result, longitudinal leptin changes in the prepubertal period demonstrated three distinguished patterns (Figure 1). They are the slow-rising, rapid-rising, and stabilized forms. Among them, the highest BMI at all time points was observed in the rapid-rising group, indicating that an elevation of leptin level in childhood could be a risk factor for obesity development in the near future. In line with this observation, Li et al. also demonstrated a similar pattern of serum leptin movement in childhood and up to adolescence at age 13 years, confirming a role for serum leptin level as a determinant for future cardiometabolic outcomes [73]. Chronic elevation in serum leptin levels during the prepubertal period tend to develop higher BMI z-scores and global metabolic risk scores, although risk for hypertension is low with a lower systolic blood pressure z-score in early adolescence, compared to those in the leptin-stable group [73]. Additional investigations further indicate a positive correlation between leptin and increased cardiometabolic risk, including obesity and hypertension, during adolescence [74,75,76]. Together, these results clearly indicate a strong relationship between serum leptin level in neonates, children, and adolescents and future cardiometabolic outcomes.

As mentioned above, levels of serum leptin are positively correlated with body adiposity and BMI in the prepubertal period. In neonates, increased adiposity is a major contributor to overall body weight gain, especially in the first 6 months [77]. Importantly, it has been suggested that faster body weight gain in early infancy increases the risk of obesity in later life. In support, a recent investigation revealed that greater body weight gain, represented by net change in BMI z-score, in the first 6 months is associated with greater adiposity and C-reactive protein, a cardiometabolic biomarker, during mid-childhood with an age between 6.6 and 10.7 years [78]. This effect was enhanced by including larger birth sizes in the correlation [78]. On the other hand, lower birth size in combination with faster body weight gain during 0.5–1 year predicted higher adiposity, leptin, and insulin resistance in mid-childhood [78]. Therefore, birth size and body weight gain during the specific developmental period should be considered together when predicting potential childhood cardiometabolic outputs. Independent investigations further recognized a strong association between infants’ BMI peak and early childhood cardiometabolic outputs [79,80]. The peak in infant BMI occurs around the age of 6 months, and both age and the magnitude of BMI peak are positively associated with BMI and blood pressure by 4 years of age, showing a delayed BMI peak with an increased risk of childhood obesity. It is worth mentioning that although serum leptin level generally represents body adiposity, this correlation is not consistent throughout life. For example, the level of cord leptin is positively related with body fat and BMI at the age of 9, but not at age 17 [46]. This finding indicates the existence of leptin-associated metabolic programming that occurs both dependently and independently of body adiposity/BMI during childhood.

### 2.3. Merge of Leptin with Early Environmental Factors

Accumulating evidence has suggested the risk of maternal smoking to the development of metabolic diseases of their offspring. For example, a link between gestational smoking and risk of offspring obesity and hypertension in childhood has been observed [81,82,83]. Nevertheless, the level of maternal and cord leptin is independent of maternal smoking in humans [84,85]. The serum leptin level of newborn infants is also comparable regardless of maternal smoking, although the birth weight of newborns from smoking mothers is significantly lower than those from non-smoking mothers [86]. The level of serum leptin was significantly elevated in 3-month-old infants of smoking mothers [86]. In rats, exposure to smoking during the neonatal period leads to the development of an obese phenotype without change in food intake [87]. Additionally, offspring with neonatal smoke exposure developed leptin resistance and impaired leptin signaling in the brain, indicating a possible link between maternal smoking (including gestational and postnatal period) and offspring central leptin signaling and metabolic diseases development in later life. Importantly, Yousefi et al. reported a possible prolonged effect of maternal smoking on genetic mutation in leptin and/or leptin receptor genes, resulting in elevated serum leptin levels and body weight in later childhood [88]. In addition to maternal effects, smoking of fathers is also a risk factor of childhood metabolic problems in association with elevated serum leptin levels [89]. Together, offspring exposed to smoking in their early life possess a high risk of cardiometabolic problems in later life, presumably through an impaired leptin mechanism including central leptin resistance. 

Similar to smoking, parental alcohol consumption is also a risk factor for metabolic impairment in their children [89]. The degree of body cholesterol and BMI of children is positively associated with increased parental alcohol consumption [89]. Importantly, alcohol drinking is a socio-additive behavior within family members, and the level of alcohol consumption of children is strongly associated with parental alcohol drinking [90]. In line with this observation, alcohol consumption in childhood, independent of parental effects, is also a strong risk factor driving metabolic abnormality in later life [90]. The association between parental or childhood alcohol drinking and leptin for future cardiometabolic programing is not yet clear. Furthermore, the effect of alcohol on leptin biology in adulthood is somewhat confusing. In both animal and human studies, the serum leptin level either increased, decreased, or was unaltered with alcohol drinking [91,92,93,94,95]. However, alcohol consumption altered the relationship between genetically determined leptin and serum lipids [96]. These investigations suggest the possible involvement of leptin signaling in an alcohol-mediated cardiometabolic programming, although a more detailed underlying mechanism of leptin determining alcohol-dependent cardiometabolic fate remains to be elucidated in the future.

## 3. Leptin as a Hallmark of Cardiometabolic Fate in Adulthood

As described previously, leptin circulates throughout the entire body after being secreted from the adipose tissue, allowing it to trigger leptin receptor-dependent multiple synaptic networks within the CNS. In adulthood, the activity of the leptin-dependent synaptic network can be initiated by the circumventricular organs [5,6,97,98,99], by the hypothalamus [2,3], or by the extra-hypothalamic deep brain regions [100,101], to modulate appetite behavior and energy expenditure [2,3]. In addition, central leptin induces SNA to regulate cardiovascular system including the baroreflex tone, heart rate, and blood pressure, and leptin-associated chronic overactivation of SNA causes the development of metabolic syndrome [102,103,104,105,106]. Importantly, the concentration of plasma leptin is directly and positively associated with the gene expression of the leptin receptor in the brain, which is achieved through the modulation of the number of cells expressing leptin receptors [107]. Thus, peripheral leptin and the central counterpart closely interact to regulate cardiometabolic homeostasis in adulthood. This also implicates abnormal serum leptin in inducing impairment in CNS leptin receptor-dependent mechanisms and contributing to metabolic syndrome development [108,109]. Therefore, the degree of serum leptin in adulthood could be a hallmark indicator for cardiometabolic fate in a later stage of life. 

In support of this view, multiple independent investigations have recognized the possibility of the level of serum leptin in adulthood as a predictor of metabolic diseases development. In middle-aged and older men (mean age between 50 and 80 years), the degree of blood leptin was positively associated with the risk of obesity onset within a few years [19,20]. In healthy middle-aged men, around 50 to 60 years, the level of serum leptin was also indicative of a risk of diabetic progress [21]. In addition, serum leptin in patients with multiple metabolic syndromes was significantly higher than in age-matched healthy men (6.7 vs. 4.6 ng/mL, respectively; [22]). Therefore, blood leptin levels are a biomarker to predict metabolic fate in adulthood. 

As previously mentioned, obesity is a foundation of cardiovascular risk. Because the level of plasma leptin is a biomarker for metabolic impairment, such as obesity and type 2 diabetes, it could also represent the risk of cardiovascular dysfunction [110,111]. In support of this conjecture, elevation in serum leptin in obese rodents, induced by a high-fat diet, developed high blood pressure through a CNS mechanism [112]. In human studies, middle-aged men with high blood leptin tended to develop obesity, as well as a high blood pressure [19]. Additionally, the level of serum leptin was higher in the obese population than in non-obese men, and elevated serum leptin was responsible for the obesity-associated increase in diastolic blood pressure [113]. On the other hand, high blood leptin can be a risk factor for cardiovascular diseases independently of obesity. In humans, a 20-year longitudinal study, starting from middle-aged healthy men aged around 40 to 60, recognized a significant positive correlation between the level of serum leptin and the risk of heart failure, independent of BMI [114]. Furthermore, serum leptin in middle-aged men was also strongly linked with obesity-independent insulin resistance and glucose intolerance, which are the risk factors that exacerbate metabolic syndrome [115]. Together, blood leptin is a risk factor for metabolic diseases, as well as for cardiovascular impairments, that occurs dependently or independently of obesity [116]. 

It is worth noting that the relationship between serum leptin and metabolic syndrome is very prominent in men and postmenopausal women, but weak in young women [117,118]. In general, serum leptin levels in women are much higher than in men, and women possess a higher obesity rate than men [61,118,119]. However, premenopausal women are protected from hypertension development, and plasma leptin during the premenopausal period is not associated with the risk of metabolic syndrome [117,120]. Thus, the strong relationship between leptin and metabolic syndrome is established after menopause [118,120], suggesting a sex-specific leptin effect on cardiometabolic physiology, with a possible interaction between leptin and female sex hormones.

## 4. Conclusion

Temporal leptin-associated physiological parameters affecting future cardiometabolic output are summarized in Table 1. The concentration of serum leptin dynamically fluctuates throughout an individual’s life, attributable to age, metabolic state, and physiological conditions. Importantly, alterations of circulating leptin predict cardiometabolic fate in the future throughout the entire life. However, a longitudinal study with sustained monitoring of leptin and cardiometabolic alterations beginning at gestation and continuing throughout life remains unexplored, and therefore, should be addressed in the future.

In the context of leptin-dependent cardiometabolic physiology, the CNS plays a critical role in producing adequate neuroendocrine and autonomic output in response to circulating leptin to balance cardiometabolic homeostasis. In this regard, numerous investigations have pointed to the hypothalamus as a central target for serum leptin. However, recent information has also recognized the possible engagement of extra-hypothalamic and non-neuronal cell populations in the brain to regulate leptin-associated cardiometabolic mechanisms [2,5,6,100,101]. Other factors, such as sex steroids, cardiovascular hormones, and neuroinflammatory molecules, have also been suggested to interact with leptin in the CNS to affect the whole body cardiometabolic system [6,121,122]. Therefore, multidirectional approaches are necessary to uncover the contribution of central leptin in determining metabolic and cardiovascular fate. Altogether, the level of circulating leptin is a strong determinant for future cardiovascular and metabolic conditions throughout the entire life.

## Figures and Tables

**Figure 1 nutrients-12-03256-f001:**
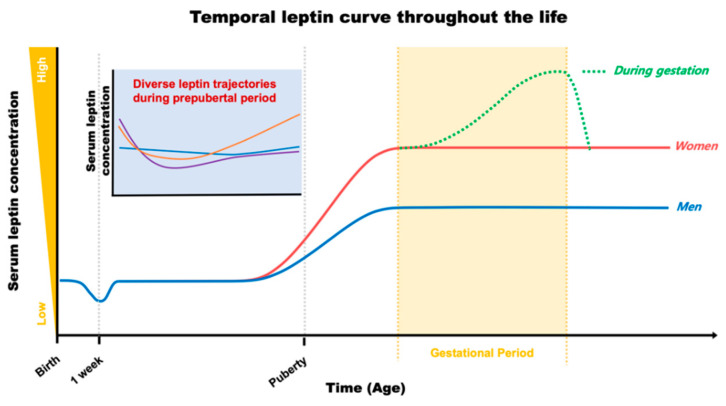
Plasma leptin trajectories throughout the life in humans.

**Table 1 nutrients-12-03256-t001:** Temporal leptin-associated physiological factors determining future cardiometabolic fate.

Time	Physiological Factors in Association with Leptin	Ref.
**Gestational Period**	Temporal leptin rhythm BMI of pregnant women Genetic modification (leptin/leptin receptor) Maternal leptin/insulin resistance	[18,23,31,32,33] [23,33] [13] [18,26,27,28]
**Neonatal to early life**	Birth size/weight BMI peak Post-natal leptin surge/rhythm Brain function Inflammation Genetic modification (leptin/leptin receptor) Insulin/Glucose homeostasis Post-natal body weight gain	[17,23,33,34,36,43] [79,80] [61,73] [65] [74] [13] [67] [44,45,46,49,50,51,77,78]
**Adulthood**	Brain function (leptin receptor pathway, SNA activity) Insulin/Glucose homeostasis Genetic modification (leptin/leptin receptor)	[102,103,104,106,107] [115] [14,15,16,107]

SNA: sympathetic nervous activity.

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
