# Peer review of "Temporal Leptin to Determine Cardiovascular and Metabolic Fate throughout the Life"

_nutrients, 2020, doi:10.3390/nu12113256_

Round 1

Reviewer 1 Report

In this manuscript Jae Geun Kim et al. evaluated the contribution of central leptin in determining metabolic and cardiovascular fate. This paper represents a fine overview of the recent papers on this topic. The literature reported used to write the paper is of good quality. Theme has been properly described. Since, the text contains a large amount of information I would suggest to the authors to increase the number of graphical figures that might help the readers.

Author Response

Dear Reviewer 1,

First of all, the authors would like to thanks Reviewer 1 for valuable comment, and we modified the manuscript following the comment as below;

Comment 1;

In this manuscript Jae Geun Kim et al. evaluated the contribution of central leptin in determining metabolic and cardiovascular fate. This paper represents a fine overview of the recent papers on this topic. The literature reported used to write the paper is of good quality. Theme has been properly described. Since, the text contains a large amount of information I would suggest to the authors to increase the number of graphical figures that might help the readers.

Answer: We agree with the comment. In addition, we believe that Figure 1 is too much simplified, and therefore, not enough to represent complex information described in the text. Thus, the authors modified Figure 1 to clearly show temporal leptin trajectories, including multiple patterns in prepubertal period as shown in a small box within the Figure 1, as well as a decrease in leptin level in first week of life. By modifying Figure 1, we believe it represents and covers most of manuscript without generating more images. We hope Review 1 also accept modified version of Figure 1 and agree with our point of view.

Reviewer 2 Report

In this manuscript, Kim et al. summarized recent updates for leptin and metabolic diseases, focusing of the temporal changes in leptin levels during pregnancy (maternal and prenatal levels), postnatal period, and adulthood. This aim is interesting and provides useful information to readers. However, I feel that the following points should be addressed before I would recommend this manuscript for publication. Generally this review is written in good English, but minor proofreading is required before publication.

General comments:
I encourage authors to include plasma leptin concentration ranges as much as possible. For example, how high is "abnormal high level of maternal leptin (L82)"? Pregnant and non-pregnant? Effect of obesity? It will be helpful for readers.

I feel that English expressions are often insufficient. I point out some below, but please have a thorough English proofreading.
L86: The authors write that hormones and amino acids are related to fetal development. It is more accurate to use "hormone levels" and "amino acid levels" here.
L88: Cord leptin levels and maternal leptin levels.
L92: It does not make sense that cord leptin predicts childhood cardiovascular and metabolic state. Should be cord leptin levels.
L119: correlated with the risk of CVD?

Please clarify whether maternal and prenatal leptin exert the metabolic effect via CNS. The first half of Abstract describes leptin's effects on the CNS, from which I had an impression that the main focus of this review is leptin in the CNS. However, CNS does not frequently appear in the main text, and is mainly explained in the section 3. i. e., Abstract and text are inconsistent.

Section 2.1
The authors describe many associations between cord leptin levels, birth weight, and postnatal parameters. Papers cited here are mostly human association studies, but are there any animal studies that suggest the causal effect of maternal or cord leptin levels on physiological parameters? This section is not very convincing if the high prenatal leptin levels are simply the consequence of high body weight. In section 3, the authors clearly describe that high leptin is a risk factor of metabolic diseases independently of obesity.

Section 2.2
While the main focus of this review is temporal leptin levels, more attention is paid to body weight.Specific comments:
L19, L41: Word usage. "Metabolic syndrome" is the name for a group of risk factors that raises the risk for some health problems including diabetes, (see this NIH website: https://www.nhlbi.nih.gov/health-topics/metabolic-syndrome) and thus does not include type 2 diabetes. "Metabolic disease" may be a better word, but it would be more accurate to simply write "...in the development of hypertension, obesity, and type 2 diabetes."

L40: Reconsider the sentence structure. It is difficult to interpret "respectively" here.
L73: Should be continued from L72?
L75: Spell out BMI at first appearance
L113: Reference needed

Author Response

Dear Reviewer 2,

First of all, the authors would like to thanks Reviewer 2 for valuable comments, and we tried to do our best to complete the manuscript following the comments as below;

Comment 2;

this manuscript, Kim et al. summarized recent updates for leptin and metabolic diseases, focusing of the temporal changes in leptin levels during pregnancy (maternal and prenatal levels), postnatal period, and adulthood. This aim is interesting and provides useful information to readers. However, I feel that the following points should be addressed before I would recommend this manuscript for publication. Generally this review is written in good English, but minor proofreading is required before publication.

General comments:
I encourage authors to include plasma leptin concentration ranges as much as possible. For example, how high is "abnormal high level of maternal leptin (L82)"? Pregnant and non-pregnant? Effect of obesity? It will be helpful for readers.

Answer: the revised manuscript has now included plasma leptin concentration during gestation time (line 72), in cord blood (line 126), in newborn infants (line 170), in 1 week old infants (line 172) and in adulthood (lines 103 and 303 for normal and patients with metabolic syndromes). As the range of serum leptin varies depend on studies, reference is noted after leptin concentration).

I feel that English expressions are often insufficient. I point out some below, but please have a thorough English proofreading.
L86: The authors write that hormones and amino acids are related to fetal development. It is more accurate to use "hormone levels" and "amino acid levels" here.
L88: Cord leptin levels and maternal leptin levels.
L92: It does not make sense that cord leptin predicts childhood cardiovascular and metabolic state. Should be cord leptin levels.
L119: correlated with the risk of CVD?

Answer: please accept our apology for inconvenient English expression. Expression in English has been corrected throughout the entire manuscript, including the areas as mentioned above.

Please clarify whether maternal and prenatal leptin exert the metabolic effect via CNS. The first half of Abstract describes leptin's effects on the CNS, from which I had an impression that the main focus of this review is leptin in the CNS. However, CNS does not frequently appear in the main text, and is mainly explained in the section 3. i. e., Abstract and text are inconsistent.

Answer: The authors agree with your view. Following the suggestion, the revised manuscript includes now more paragraphs to deal with the CNS and leptin; lines 78-83 and 107-118.

Section 2.1
The authors describe many associations between cord leptin levels, birth weight, and postnatal parameters. Papers cited here are mostly human association studies, but are there any animal studies that suggest the causal effect of maternal or cord leptin levels on physiological parameters? This section is not very convincing if the high prenatal leptin levels are simply the consequence of high body weight. In section 3, the authors clearly describe that high leptin is a risk factor of metabolic diseases independently of obesity.

Answer: The authors agree with the comments, and included more information from animal studies in the revised manuscript. Lines 78-83 and 107-118. The authors also described possible leptin actions, dependently and independently of BMI/adiposity in childhood (lines 220-224).

Section 2.2
While the main focus of this review is temporal leptin levels, more attention is paid to body weight.

Answer: The authors agree with the comment, and modified the text to make stronger leptin story (lines 183-202).Specific comments:
L19, L41: Word usage. "Metabolic syndrome" is the name for a group of risk factors that raises the risk for some health problems including diabetes, (see this NIH website: https://www.nhlbi.nih.gov/health-topics/metabolic-syndrome) and thus does not include type 2 diabetes. "Metabolic disease" may be a better word, but it would be more accurate to simply write "...in the development of hypertension, obesity, and type 2 diabetes."

Answer: The authors thanks for the comment, and “Metabolic syndrome” has been changed to “Metabolic diseases” throughout the entire manuscript.

L40: Reconsider the sentence structure. It is difficult to interpret "respectively" here.
L73: Should be continued from L72?
L75: Spell out BMI at first appearance
L113: Reference needed

Answer: The authors thanks for the comments, and they are all modified/corrected now in the revised manuscript.

Reviewer 3 Report

I consider the article entitled: Temporal leptin to determine cardiovascular and metabolic fate throughout the life by Jae Geun Kim, Byung Ju Lee, Jin Kwon Jeong to be very interesting, especially in the context of the topic. The article has been prepared correctly and is easy to read, that is why I believe that it should be seriously considered for publication after adding some information.
It seems to me that their lack may weaken this good article.

I think that the authors in their work should consider three issues:

  1. Leptin-resistance.
  2. The existence of three distinct leptin trajectories from birth to mid-childhood. Such trajectories are low-stable, high-decreasing and intermediate-increasing. The intermediate-increasing trajectory was associated with early teen cardio-metabolic risk. I believe the authors of this article should read the article entitled: Leptin Trajectories from Birth to Mid-Childhood and Cardio-Metabolic Health in Early Adolescence.
    Other researchers have also written about such trajectories before, e.g. Volberg V at al., (2013), Gruszfeld at al., in the CHAMACOS and CHOP studies

In this context, perhaps authors of the reviewed article should rethink the appearance of figure 1 (which in the pre-pubertal period)  seems to be a bit too simplistic.

It should be pointed out that the authors of the previously mentioned articles went a step further and tried to estimate associations between distinctive leptin trajectories and metabolic risk scores.

  1. Maybe the article should be completed with the influence of factors such as mothers' smoking during pregnancy, maternal overweight, and formula feeding different distinguished courses of leptin trajectories.

To sum up, the article is engaging, stylistically well-written, but too simplified in some places and should be corrected with this additional issue.

Best regards

Author Response

Dear Reviewer 3,

First of all, the authors would like to thanks Reviewer 3 for valuable comments, and the revised manuscript has been modified/corrected following the comments below;

Comment 3;

I consider the article entitled: Temporal leptin to determine cardiovascular and metabolic fate throughout the life by Jae Geun Kim, Byung Ju Lee, Jin Kwon Jeong to be very interesting, especially in the context of the topic. The article has been prepared correctly and is easy to read, that is why I believe that it should be seriously considered for publication after adding some information.
It seems to me that their lack may weaken this good article.

I think that the authors in their work should consider three issues:

  1. Leptin-resistance.

Answer: The authors focused here a normal (and abnormal) movement of life time serum leptin concentration, and function to predict cardiometabolic output in later life. With metabolic diseases, serum leptin level stays high with a lower leptin receptor responsiveness, which is leptin resistance. Although it is still uncertain whether leptin resistance causes metabolic disorders or metabolic diseases develop leptin resistance, leptin resistance is clearly associated with metabolic dysfunction. The revised manuscript has now included a contribution of leptin resistance for leptin-associated cardiometabolic fate (lines 107-118). In addition, a term “leptin resistance” and it’s possible role in cardiometabolic programming is clear noted throughout the text to highlight it.

  1. The existence of three distinct leptin trajectories from birth to mid-childhood. Such trajectories are low-stable, high-decreasing and intermediate-increasing. The intermediate-increasing trajectory was associated with early teen cardio-metabolic risk. I believe the authors of this article should read the article entitled: Leptin Trajectories from Birth to Mid-Childhood and Cardio-Metabolic Health in Early Adolescence.
    Other researchers have also written about such trajectories before, e.g. Volberg V at al., (2013), Gruszfeld at al., in the CHAMACOS and CHOP studies

Answeer: The authors thanks to the valuable comment. Following the suggestion, the Section 2.2 of revised manuscript is modified (lines 183-202).

In this context, perhaps authors of the reviewed article should rethink the appearance of figure 1 (which in the pre-pubertal period)  seems to be a bit too simplistic.

Answer: The authors agree with the point. Figure 1 is modified to include more information, including leptin trajectories in pre-pubertal period.

It should be pointed out that the authors of the previously mentioned articles went a step further and tried to estimate associations between distinctive leptin trajectories and metabolic risk scores.

Answer: It is now described in the revised text (lines 183-202).

  1. Maybe the article should be completed with the influence of factors such as mothers' smoking during pregnancy, maternal overweight, and formula feeding different distinguished courses of leptin trajectories.

Answer: Following the suggestion, the authors included Section 2.3 in the revised version, subtitled as “Merge of leptin with early environmental factors” (lines 226-280). Here, the authors focused the association between leptin and environmental factors in metabolic programming. Especially, smoking and alcohol drinking both are a socio-additive behavior, and offspring of smoking/alcohol drinking parent are continuously exposed to smoke and learn alcohol drinking in early time, have a high risk of metabolic disorders development. Therefore, section 2.3 focused these socio-additive environmental factors to offspring metabolic outcomes in relation with leptin signaling. To be focused and make a story consistent, the authors excluded formular feeding/maternal overweight, and hope reviewer accept this.

Round 2

Reviewer 2 Report

The authors appropriately addressed all of my concerns. It was my pleasure to have reviewed this manuscript.

Author Response

Dear Reviewer.

It was our honor to have your review effort to make the manuscript better, and the authors would like thanks to you for your valuable time and support.

Stay safe from the pandemic,

Jin.